# Grapevine Trunk Diseases in Greece: Disease Incidence and Fungi Involved in Discrete Geographical Zones and Varieties

**DOI:** 10.3390/jof10010002

**Published:** 2023-12-20

**Authors:** Stefanos I. Testempasis, Emmanouil A. Markakis, Georgia I. Tavlaki, Stefanos K. Soultatos, Christos Tsoukas, Danai Gkizi, Aliki K. Tzima, Epameinondas Paplomatas, Georgios S. Karaoglanidis

**Affiliations:** 1Laboratory of Plant Pathology, School of Agriculture, Faculty of Agriculture, Forestry and Natural Environment, Aristotle University of Thessaloniki, 54124 Thessaloniki, Greece; testempa@agro.auth.gr (S.I.T.); gkarao@agro.auth.gr (G.S.K.); 2Laboratory of Mycology, Department of Viticulture, Vegetable Crops, Floriculture and Plant Protection, Institute of Olive Tree, Subtropical Crops and Viticulture, Hellenic Agricultural Organization—DIMITRA, 32^A^ Kastorias Street, Mesa Katsabas, 71307 Heraklion, Greece; gogotavlaki@hotmail.gr (G.I.T.); stefanoss1313@gmail.com (S.K.S.); 3Laboratory of Plant Pathology, Department of Agriculture, School of Agricultural Sciences, Hellenic Mediterranean University, Stavromenos, 71004 Heraklion, Greece; 4Laboratory of Plant Pathology, Department of Crop Science, School of Plant Sciences, Agricultural University of Athens, Iera Odos 75, Votanikos, 11855 Athens, Greece; ctsoukas@aua.gr (C.T.); aliki@aua.gr (A.K.T.); epaplom@aua.gr (E.P.); 5Department of Wine, Vine and Beverage Sciences, University of West Attica, Ag. Spyridonos 28, 12243 Athens, Greece; dgkizi@uniwa.gr

**Keywords:** *Kalmusia variispora*, *Neosetophoma italica*, *Paraconiothyrium variabile*, *Seimatosporium vitis*, meteorological conditions, wood inhabitant

## Abstract

A three-year survey was conducted to estimate the incidence of grapevine trunk diseases (GTDs) in Greece and identify fungi associated with the disease complex. In total, 310 vineyards in different geographical regions in northern, central, and southern Greece were surveyed, and 533 fungal strains were isolated from diseased vines. Morphological, physiological and molecular (5.8S rRNA gene-ITS sequencing) analyses revealed that isolates belonged to 35 distinct fungal genera, including well-known (e.g., *Botryosphaeria* sp., *Diaporthe* spp., *Eutypa* sp., *Diplodia* sp., *Fomitiporia* sp., *Phaeoacremonium* spp., *Phaeomoniella* sp.) and lesser-known (e.g., *Neosetophoma* sp., *Seimatosporium* sp., *Didymosphaeria* sp., *Kalmusia* sp.) grapevine wood inhabitants. The GTDs-inducing population structure differed significantly among the discrete geographical zones. *Phaeomoniella chlamydospora* (26.62%, *n* = 70), *Diaporthe* spp. (18.25%, *n* = 48) and *F. mediterranea* (10.27%, *n* = 27) were the most prevalent in Heraklion, whereas *D. seriata*, *Alternaria* spp., *P. chlamydospora* and *Fusarium* spp. were predominant in Nemea (central Greece). In Amyntaio and Kavala (northern Greece), *D. seriata* was the most frequently isolated species (>50% frequency). Multi-genes (*rDNA-ITS*, *LSU*, *tef1-α*, *tub2*, *act*) sequencing of selected isolates, followed by pathogenicity tests, revealed that *Neosetophoma italica*, *Seimatosporium vitis*, *Didymosphaeria variabile* and *Kalmusia variispora* caused wood infection, with the former being the most virulent. To the best of our knowledge, this is the first report of *N. italica* associated with GTDs worldwide. This is also the first record of *K. variispora*, *S. vitis* and *D. variabile* associated with wood infection of grapevine in Greece. The potential associations of disease indices with vine age, cultivar, GTD-associated population structure and the prevailing meteorological conditions in different viticultural zones in Greece are presented and discussed.

## 1. Introduction

Grapevine (*Vitis vinifera* L.) is one of the most commonly cultivated plant species in several countries like Australia, France, Greece, Italy, South Africa, Spain, and the USA, and has a significant social, economic, and environmental impact [1]. Greece belongs to the top 20 leading grapevine-growing countries of the world in terms of production quantity [2], and viticulture is essential for the economy and rural development of the country, covering an area of 89,230 ha with an annual yield of 818,860 tons in 2021 [3].

In recent decades, grapevine trunk diseases (GTDs) have emerged as the most severe threat to viticulture sustainability worldwide [4]. GTDs are a cluster of diseases incited by a vast number of fungal pathogens (mostly ascomycetes) that colonize and infect wood tissues, causing internal symptoms such as wood streaking and necrosis, vascular browning, and white rot [5,6]. Consequential external symptoms include leaf chlorosis, yellowing and wilting, wood cankers, and dieback of spurs, canes, and cordons, leading to stunted plant growth, decline, reduced longevity and productivity, and occasionally plant death.

The increased incidence of GTDs globally has been attributed to several factors like the withdrawal of effective chemicals (i.e., sodium arsenate, benzimidazole fungicides), alterations in cultivation practices, globalization in the dissemination of plant material contaminated with latent pathogens, and climate change [7,8,9,10]. In particular, climate change and global warming can increase plant stress and make vines vulnerable to various biotic stresses, including GTD-inciting pathogens, in a synergistically interactive manner [10,11]. On the other hand, rapid alterations in wet and dry cycles along with extremely low and/or high temperatures, as a consequence of climate change, may favor pathogen dispersal from their original hosts and adaption to *Vitis*, resulting in the emergence of new GTD-associated fungal species as has been speculated for other woody hosts [8,12,13]. This possibly explains the increased number of fungal pathogens identified as causal agents of GTDs nowadays compared to the past. Additionally, the availability of high-resolution molecular tools has also provided a deeper insight into the GTD-associated microbiome in recent years compared to previous ones and has enriched, so far, our knowledge of microbial species that contribute to the GTD complex [14,15,16,17,18].

To date, more than 140 fungal species belonging to 35 genera have been involved in GTDs’ complex worldwide, although the pathogenic potential has not been confirmed for all of them in pathogenicity trials [7,19,20,21]. Interestingly, recent surveys conducted in marginal temperate regions in Cyprus and southern Italy revealed several lesser-known and/or novel species associated with GTDs, like *Cadophora luteo-olivacaea*, *Colletotrichum fioriniae*, *Seimatosporium vitis-vinifera*, *Seim. cyprium* sp. nov., *Sporocadus kurdistanicus*, *Spo. rosigena* and *Truncatella angustata*, and proved their pathogenic potential [20,21].

Spatial determination of the phytopathogenic status of GTDs in grapevine-growing countries following a regional-scale mapping is important for planning and implementing effective management practices targeting specific diseases. Furthermore, the accurate estimation of disease parameters in distinct viticultural zones may provide valuable information on the susceptibility level for widely used genotypes and, thus, improved disease management through risk assessment [13].

Although GTDs’ incidence and implicated microorganisms have extensively been investigated for most grapevine-growing countries, only crude estimations have been conducted in Greece. In particular, Rumbos and Rumbou [22], determined specific fungal genera (i.e., *Phaeomoniella*, *Fomitiporia*, *Stereum*, *Phaeoacremonium*, *Cylindrocarpon*, and *Botryosphaeria*) isolated from different-aged vine samples in Greece following a culture-dependent and morphological identification approach. Due to their low isolation frequency, authors incorrectly suggested that these fungi could not be the cause of young grapevine decline. In a recent study, Bekris et al. [14] determined the fungal and bacterial microbiome in symptomatic and asymptomatic grapevines of three important cultivars grown in geographically distinct viticultural zones in Greece following a metagenomic approach and revealed several lesser-known fungal genera related to GTD-affected vines. Both studies are inadequately informative about the incidence and severity of GTDs as well as the phytopathogenic role of certain fungal species identified in the GTD-associated microbiome in the country. Therefore, the main objectives of the present study were as follows: (i) to estimate disease parameters and identify fungi associated with GTDs in Greece following a culture-dependent approach, (ii) to determine the population structure of GTDs-inciting fungi in distinct viticultural zones, (iii) to investigate the phytopathogenic potential of lesser-known fungal species on grapevine, and (iv) to identify the potential associations of disease indices with vine age, genotype and the prevailing meteorological conditions in discrete viticultural zones.

## 2. Materials and Methods

### 2.1. Field Surveys and Disease Incidence Assessment

In total, 310 vineyards in three (3) geographically distinct viticultural zones in northern (*n* = 234), central (*n* = 28), and southern (*n* = 48) Greece were surveyed during July and August of the years 2018–2020. The number of vineyards surveyed for the assessment of GTDs incidence in each grapevine-growing region and their age scale are shown in Table 1. The geographical distribution of the surveyed locations is shown in Appendix A. For each vineyard, either all or at least 100 vines in randomly selected lines in the vineyard were surveyed, and the percentage of diseased vines (including dead and non-dead vines), symptomatic but non-dead vines, and dead/apoplectic vines, was measured. Additional information about each vineyard’s age, genotype (cultivar/rootstock), training–pruning system, and geographical position was also recorded.

### 2.2. Sampling and Fungal Isolation

During the surveys (in July and August of the years 2018–2020), wood samples from 41 vineyards in southern (Heraklion), 16 vineyards in central (Nemea), and 19 vineyards in northern Greece (9 vineyards in Amyntaio and 10 vineyards in Kavala) were collected and analyzed. In each vineyard, 3 to 9 representative grapevines with typical GTDs symptoms were sampled. Overall, 252 vines were sampled from northern (*n* = 60), central (*n* = 69), and southern (*n* = 129) Greece. Wood samples from the trunk and the cordon of each vine (approximately 30 cm in length each) were cut and visually inspected to verify the occurrence of internal wood symptoms. The samples were transferred to the laboratory and cut crosswise and lengthwise, the phloem was removed, and wood fragments (approx. 10 cm in length) were surface sterilized by soaking them in 93% ethanol and passing them through a flame, thrice. Xylem chips taken from symptomatic wood tissue were aseptically placed onto acidified potato dextrose agar (PDA, Merck, Darmstadt, Germany). The plates were incubated at 24 ± 0.2 °C in the dark and observed daily for at least two weeks. The emerging fungal colonies that grew out of tissue excisions were examined visually and under a light microscope, and the fungi were transferred onto new PDA plates. For short-term storage, all fungal isolates were maintained on PDA at 4 °C, whereas for long-term storage, they were stored at −80 °C in a 25% (*v*/*v*) aqueous glycerol solution.

### 2.3. Morphological and Cultural Characterization of Isolates

To estimate the mycelial growth of fungal isolates, mycelial-colonized PDA agar discs (5 mm in diameter) was transferred into the center of new PDA plates with a diameter of 92 mm (one disc per plate, three per isolate). The plates were incubated at 24 °C in the dark, and the colonies’ diameter was measured periodically for up to three weeks or stopped earlier when the fungus completely covered the plate surface. The growth rate of fungal isolates was expressed in mm/day. At the end of the incubation period, colony characteristics (color, mycelium, colony texture, and shape) were observed. Dimensions of available conidia (30 readings per isolate), reproductive structures, and hyphal features (color, shape, presence or absence of septum, clump, and chlamydospores) were also recorded.

### 2.4. DNA Extraction, PCR Amplification and Sequencing

Molecular identification of fungal isolates was carried out by extracting fungal DNA and sequencing their internal transcribed spacer regions of ribosomal DNA (*rDNA-ITS*) gene. In addition, rDNA large subunit (*LSU*), translation elongation factor 1-alpha (*tef1-α*), β-tubulin (*tub2*) and actin (*act*) genes of isolates HOURD2.1AVR1, PEROG2.1YP2, SAROG1.3AVR10, SAROG1.3AVR13, SAROG1.2AKO1, and SAROG1.3AVR7 were sequenced. Fungal DNA was extracted from 100 mg mycelium (fresh weight), and scraped from the surface of 2-to-3-week-old PDA cultures by using a sterilized scalpel, according to the work of Cary et al. [23], with slight modifications. In brief, 100 mg mycelium per sample was transferred in a 1.5 mL microfuge tube and crushed with a sterilized pestle in the presence of 700 μL LETS buffer (0.8 mM EDTA, 0.0125% SDS, 0.1 mM Tris-HCl, 2.5 mM LiCl). The sample was mixed via inversion and incubated at room temperature for 5 min. Then, 700 μL phenol/chloroform/isoamyl alcohol (25:24:1) was added, mixed via inversion, incubated for 5 min at room temperature and centrifuged at 13,000× *g* at 4 °C for 10 min. The supernatant (approx. 500 μL) was transferred to a new tube, and 1 mL 95% EtOH was added, vortexed and centrifuged at 13,000× g at 4 °C for 10 min. The supernatant was discarded and the pellet was washed with 500 μL 70% EtOH and allowed to dry at room temperature for 5 min. The pellet was resuspended in 40 μL of 10 mM Tris-HCl (pH = 8) buffer, treated with 2 μL RNase A (5 mg/mL stock, Macherey-Nagel, GmbH & Co., KG, Duren, Germany) and incubated at 50 °C for 15 min. The quantity and quality of the obtained DNAs were determined using a Q5000 UV-Vis Spectrophotometer (Quawell, San Jose, CA, USA). The final DNA concentration of each isolate was adjusted to 20 ng ml^−1^ and stored at −20 °C until use. The primer pairs used were ITS1/ITS4 for the *rDNA-ITS* region, LROR/Un-Lo28S1220 for *LSU* [24,25], EF1-728F/EF1-986R for *tef1-α* [26], T1/Bt2b-R for *tub2* [26,27] and ACT-512F/ACT-783R for *act* [26] regions of the six isolates mentioned above. All PCR assays were carried out in an FG-TC01 FastGene^®^ Gradient thermocycler (NIPPON Genetics EUROPE) using BK 1003 KAPA Taq PCR kit (KAPABIOSYSTEMS, Wilmington, MA, USA). PCR assay for *rDNA-ITS* included an initial denaturation at 95 °C for 3 min; followed by 35 cycles of 30 s of denaturation at 94 °C, 30 s of annealing at 54 °C, and 50 s of extension at 72 °C; and a final extension step at 72 °C for 5 min. LSU amplification included initial denaturation at 95 °C for 3 min; followed by 35 cycles of 30 s of denaturation at 94 °C, 30 s of annealing at 50 °C, and 1 min of extension at 72 °C; and a final extension step at 72 °C for 10 min. Cycling conditions for *tef1-α*, *tub2* and *act* were similar to those of *rDNA-ITS* but with annealing temperatures of 53 °C, 51 °C, and 57 °C, respectively. Amplified products were purified with the “NuncleoSpin^®^ Gel and PCR Clean-up” kit (MACHEREY-NAGEL, Düren, Germany) and sequenced using both forward and reverse primers at Macrogen Europe B.V., Amsterdam, the Netherlands. The “BioEdit 7.0.1” software was used to edit the raw sequencing data [28]. For isolates HOURD2.1AVR1, PEROG2.1YP2, SAROG1.3AVR10, SAROG1.3AVR13, SAROG1.2AKO1, and SAROG1.3AVR7, the assembled sequences of their *rDNA-ITS*, *LSU*, *tef1-α*, *tub2*, and *act* regions were deposited in GenBank under the accession numbers shown in Appendix A.

### 2.5. Identification and Characterization of Isolates

The fungal isolates obtained from diseased wood tissues were characterized using morphological and physiological features, along with *rDNA-ITS* gene sequences. Molecular identification of the wood-inhabiting fungi was conducted by utilizing the “blastn” option at NCBI and comparing their gene sequences with those in the GenBank database. Additionally, for six (6) selected lesser-known fungal isolates (HOURD2.1AVR1, PEROG2.1YP2, SAROG1.3AVR10, SAROG1.3AVR13, SAROG1.2AKO1, and SAROG1.3AVR7), their *LSU*, *tef1-α*, *tub2*, and *act* gene sequences were employed. Phylogenetic analysis was performed to assess the relationships of these isolates within the genera *Neosetophoma*, *Didymosphaeria*, *Seimatosporium*, and *Kalmusia* (Appendix A). Phylogenetic analysis was carried out for each respective genus using two or more genomic regions: *Seimatosporium* (*rDNA-ITS*, *LSU*, *tef1-α*, and *tub2*), *Didymosphaeria* (*rDNA-ITS*, *LSU*, *tub2*), *Kalmusia*, and *Neosetophoma* (*rDNA-ITS*, *LSU*). A combined dataset of aligned multi-locus sequences was constructed using “ClustalW version 1.81”, and evolutionary analyses was conducted in Geneious Prime^®^ 2022.1.1 software by employing the Neighbor-joining method (NJ) and the Tamura–Nei model [29]. The NJ consensus trees were based on 1000 bootstrap replications.

### 2.6. Correlation between Meteorological Data and GTDs Pathogens’ Frequency

The potential associations of GTD-incited population structure with the prevailing meteorological conditions in different viticultural zones in Greece were investigated by inquiring and obtaining meteorological data over the last decade (2010–2020) in the regions of interest from the Hellenic National Meteorological Service. In detail, a Principle Component Analysis (PCA) was conducted to assess the correlation between the frequency of GTD pathogens and endophytic fungi isolated from each region and the average values of air temperature, relative humidity, and rainfall prevailed. PCA analysis was performed using GraphPad Prism Software Version 10.0.0 for Windows, Boston, MA, USA.

### 2.7. Pathogenicity Tests

In April 2022, a field trial was set up to confirm the pathogenicity of four isolates, representative of *Seimatosporium vitis* (isolate SARO1.3AVR10), *Didymosphaeria variabile* (isolate SARO1.2AKO1), *Kalmusia variispora* (isolate HOURD2.1AVR1) and *Neosetophoma italica* (isolate SARO1.3AVR7), identified via morphological and molecular analyses. The former three species have not been previously identified as causal agents of GTDs in Greece, whereas the latter species has never been reported as infecting grapevine worldwide. Isolates were used to inoculate 2-year-old canes of the vine cv. Soultanina grafted onto the 110 Richter rootstock. Pathogenicity trials were conducted in a 15-year-old vineyard located in Hellenic Mediterranean University, Crete, southern Greece. Only vines that remained visibly healthy for at least the last 3 years were included in the experiment.

Artificial inoculation was performed, according to Markakis et al. [12,13]. In brief, a hole measuring 6.0 mm in diameter and 10.0 mm in length was opened into the cane using a Black & Decker EPC 12 drill, and one 5 mm diameter mycelial disc taken from a 2-week-old PDA culture was inserted into the hole. Then, the hole was sealed with cellophane membrane and covered with adhesive paper tape to protect the inoculum. In total, twenty canes (ten vines with two canes each) were inoculated with each isolate. Additionally, a set of twenty canes (ten vines with two canes each) were similarly treated with sterilized PDA discs and served as controls. Treated canes remained under ambient conditions and were inspected periodically for foliar symptom development.

All the inoculated and control canes were collected in October 2022 (6 months post-inoculation), their leaves were removed, and longitudinal and transverse sections were performed to estimate the extension of wood tissue symptoms above and below the inoculation point. Five representative canes per inoculated fungus were randomly selected to verify the presence of the applied fungi in wood tissues, and pathogen re-isolations onto APDA were conducted as mentioned above.

### 2.8. Statistical Analysis

The data were checked for normality of distribution using the Kolmogorov–Smirnov and Shapiro–Wilk tests, and for homogeneity using Levene’s test. The data on disease indices in the field surveys were analyzed using the Kruskal–Wallis (non-parametric) test, followed by Dunn’s post hoc multiple comparison test of average rankings (*p* ≤ 0.05). The percentage of vines and vineyards infected by individual fungal genera in three viticultural zones were analyzed using the Chi-square test via pair-wise comparisons (at *p* ≤ 0.05). In pathogenicity trials, analysis of variance (ANOVA) was employed to evaluate the pathogenic effects of the fungal strains on wood discoloration length in canes. When a significant *F* test was obtained for treatments (*p* ≤ 0.05), the data were subjected to means separation via Tukey’s HSD test. Standards errors of means were also calculated.

## 3. Results

### 3.1. Disease Incidence in Respect to Geographical Region, Vine Age and Cultivar

Vineyards in central and southern Greece showed significantly higher (*p* < 0.05) frequencies of diseased and symptomatic/non-dead vines than those in northern Greece (Figure 1). In contrast, a significantly higher (*p* < 0.05) percentage of dead/apoplectic vines was recorded in northern than in southern Greece, whereas no significant difference (*p* > 0.05) was observed between central and northern and between central and southern Greece in terms of dead/apoplectic vines (Figure 1).

In respect to the vineyard age, elder vines (≥20 years old) showed significantly higher (*p* < 0.05) disease incidence than the middle-aged (10–19 years old) vines, whereas younger vines demonstrated significantly lower (*p* < 0.05) disease incidence compared to the ones belonging to the other two age categories (Figure 2).

The frequencies of symptomatic/non-dead plants in vineyards of Heraklion region (southern Greece) and Nemea (central Greece) were significantly higher (*p* < 0.05) than those of Amyntaio, Kavala, Lemnos, Naousa and Nea Mesimvria (northern Greece) (Figure 3). However, no significant difference (*p* > 0.05) in the percentage of dead/apoplectic vines was found among the studied regions (Figure 3).

Different grapevine cultivars varied significantly in terms of diseased (df = 23, *x*^2^ = 64.092, *p* < 0.001), symptomatic/non-dead (df = 23, *x*^2^ = 70.420, *p* < 0.001), and dead/apoplectic vines (df = 23, *x*^2^ = 44.575, *p* ≤ 0.001) frequencies. In detail, the frequency of diseased vines of cvs. Moschofilero, Tzortzina, Nebiollo, Liatiko and Asyrtiko was significantly lower (*p* < 0.05) compared to Cabernet Sauvignon, Traminer, Vidiano, Tempranillo, Soultanina, Agiorgitiko, Cisnaut, and Kotsifali. At the same time, most of the rest cultivars did not differ significantly (*p* > 0.05) from the highly or the lowly affected cultivars mentioned above (Figure 4A). Moreover, the frequency of symptomatic/non-dead vines in Soultanina, Agiorgitiko and Kotsifali vineyards was significantly higher (*p* < 0.05) than those of most of the other cultivars tested (Figure 4B). Furthermore, the frequency of dead-apoplectic vines of Moschofilero was significantly lower (*p* < 0.05) compared to that of Syrah, Tempranillo, Agiorgitiko, and Cinsaut, whereas most of the other cultivars did not differ significantly (*p* > 0.05) from the above cultivar groups in terms of dead/apoplectic vines’ frequency (Figure 4C).

### 3.2. Fungal Genera Frequencies

In total, 533 isolates were obtained from northern (*n* = 124), central (*n* = 146) and southern (*n* = 263) Greece. The determination of fungal genera based on morphological and molecular (5.8S rRNA gene-ITS sequencing) analyses revealed that isolates belonged to 35 distinct fungal genera (Figure 5). Amongst them, 18 fungal genera (including 407 out of 533 isolates) have been commonly associated with GTDs worldwide, including well-known (i.e., *Botryosphaeria dothidea*, *Cytospora* spp., *Diaporthe* spp., *Diplodia* sp., *Dothiorella* sp., *Eutypa* sp., *Eutypella* sp., *Fomitiporia mediterranea*, *Ilyonectria* sp., *Neofusicoccum* sp., *Phaeoacremonium* spp., *Phaeomoniella clamydospora*, *Phellinus* sp.) and lesser-known (i.e., *Didymosphaeria* sp., *Fusarium* sp., *Seimatosporium* sp., *Kalmusia* sp.) grapevine wood inhabitants, whereas the remaining 17 determined genera (e.g., *Alternaria* sp., *Aspergillus* sp., *Clonostachys* sp., *Neosetophoma* sp., *Trichoderma* sp., etc.) have never/rarely been associated with GTDs.

The GTDs-inducing population structure differed significantly (*p* < 0.05) among the discrete geographical zones. *Phaeomoniella chlamydospora* (26.62%, *n* = 70/263), *Diaporthe* spp. (18.25%, *n* = 48/263) and *F. mediterranea* (10.27%, *n* = 27/263) were the most prevalent in Heraklion, whereas *D. seriata* (28.77%, *n* = 42/146), *P. chlamydospora* and *Fusarium* spp. (10.27%, *n* = 15/146) were predominant in Nemea (central Greece). In Amyntaio and Kavala (northern Greece), *D. seriata* was the most frequently isolated species (51.61%, *n* = 64/124), followed by *Lasiodiplodia* sp. (8.87%, *n* = 11/124). Interestingly, *P. chlamydospora* was not isolated from vines in northern Greece, nor was *Lasiodiplodia* sp. from southern or central Greece, and *Diaporthe* sp. was not found in central Greece (Figure 5).

Overall, increased frequencies of isolates belonging to GTD-associated genera originated from elder (39.80%, *n* = 162/407) and middle-aged (49.39%, *n* = 201/407) vines compared to the younger ones (10.81%, *n* = 44/407). The relative frequencies of isolates per fungal genus and vine age category are shown in Figure 6. Apart from *Botryosphaeria* sp. and *Fusarium* sp., isolates of the most frequently occurring genera were obtained mainly from middle-aged and elder vines. Although at low frequencies, 11 out of the 18 most commonly known GTD-associated genera were also found in younger vines.

### 3.3. Incidence of GTD-Associated Fungal Genera in Vineyards and Vines in Discrete Viticultural Zones

The incidence of specific fungal genera associated with GTDs in vineyards and vines in discrete viticultural zones differed significantly (*p* < 0.05). The percentage of vineyards in northern (Amyntaio and Kavala), central (Nemea), and southern (Heraklion) Greece, where each of the 18 commonly known GTD-associated genera was detected, is shown in Figure 7.

In total, 10 out of 18 genera were found at significantly different (*p* < 0.05) frequencies among the vineyards of the discrete zones. Furthermore, nine (9) genera were isolated at significantly different (*p* < 0.05) frequencies among the vines of discrete zones (Figure 8). Indicatively, the percentage of diseased vines that were infected by *Diplodia* spp. (mostly *D. seriata*) in northern Greece (58.33%) was significantly higher (*p* < 0.05) than that in central Greece (42.03%), while the percentage of *Diplodia*-infected vines in southern Greece (4.88%) was significantly lower (*p* < 0.05) compared to that in the other two zones. Contrariwise, the percentage of vines that were infected by *Phaeomoniella* sp. (*P. chlamydospora*) in southern Greece (28.46%) was significantly higher (*p* < 0.05) than that in central Greece (8.70%), whereas no *Phaeomoniella*-infected vines were found in northern Greece (0.00%) (Figure 8). Furthermore, 26.32% of the studied vineyards and 10.00% of the symptomatic vines were infected by *Lasiodiplodia* spp. (mostly *L. viticola* and *L. theobromae*) in northern Greece, exclusively (Figure 7 and Figure 8).

### 3.4. Correlation between Meteorological Data and GTDs Pathogens’ Frequency

PCA analysis allowed us to investigate the possible correlation of the past decade’s meteorological data (air temperature, rainfall, and relative humidity) with the frequencies of the prevalent fungal genera isolated from symptomatic grapevines of Southern, Central, and Northern Greece (Figure 9). PCA analysis was performed on the correlation matrix of 23 variables, and the first two components explained 100% of the total variation in the data (68.85% for PC1, and 31.15% for PC2) (Figure 9). The arrangement of variables on the graph reveals how they are clustered together across the two dimensions. More specifically, the three examined viticultural zones were differentiated into three clusters; PC2 separated the viticultural zones of northern and central Greece, whereas PC1 only differentiated the datasets representing the southern zone (Figure 9). The correlation between two variables is determined by the angle formed by the arrows pointing towards them; thus, an acute angle indicates a (strong) positive correlation, a right angle represents no correlation, and an obtuse angle signifies a (strong) negative correlation. Furthermore, arrows pointing in the same direction indicate a positive correlation, and the length of the arrow reflects the strength of the correlation pattern. Conversely, arrows pointing in opposite directions reveal a negative correlation. A positive correlation was noticed among the southern zone of Greece, air temperature, relative humidity (RH), and several endophytic fungal genera, such as *Phaemoniella* sp., *Phaeoacremonium* sp., *Fomitiporia* sp., *Seimatosporium* sp., *Botryosphaeria* sp., *Kalmusia* sp., *Diaporthe* sp., and *Didymosphaeria* sp. In contrast, the fungal genera of *Eutypa* sp., *Neofusicoccum* sp., *Diplodia* sp., and *Lasiodiplodia* sp. were positively correlated with the northern zones of Greece and negatively correlated with the temperature (Figure 9). Additionally, the central zone of Greece was positively correlated with rainfall and the fungal genera of *Phellinus* sp., *Aspergillus* sp., *Cytospora* sp., *Fusarium* sp., and *Alternaria* sp. (Figure 9).

### 3.5. Morphological, Physiological, and Molecular Identification of Lesser-Known Fungal Species

Morphological and physiological characteristics of the six (6) selected isolates (HOURD2.1AVR1, PEROG2.1YP2, SAROG1.3AVR10, SAROG1.3AVR13, SAROG1.2AKO1, and SAROG1.3AVR7), along with their GenBank accession numbers and identities based on the analysis of their *rDNA-ITS*, *LSU*, *tef1-α*, *tub2* and *act* sequences, are presented in Appendix A. These isolates were selected and analyzed thoroughly since they belonged to lesser-known GTD-associated fungal genera that have not been previously reported in Greece (i.e., *Seimatosporium*, *Didymosphaeria* and *Kalmusia*) or genera that have not been reported as GTD-inciting agents worldwide (i.e., *Neosetophoma*). Apart from the *tef1-α* sequence of the isolate PEROG2.1YP2 (*K. variispora*), all isolates showed over 99.50% similarity with already published *rDNA-ITS*, *LSU*, *tef1-α* and *tub2* sequences that are deposited in NCBI. Moreover, gene sequences of the fungal species that were not available in NCBI were deposited (i.e., *act* sequences of *K. variispora* and *S. vitis*, *LSU* sequence of *D. variabile*, and *LSU*, *tef1-α*, *tub2* and *act* sequences of *N. italica*).

In this study, we conducted a comprehensive phylogenetic analysis using combined alignments of *rDNA-ITS, LSU, tub2*, and *tef1-a* sequences from two *S. vitis* isolates (SAROG1.3AVR10 and SAROG1.3AVR13). The Neighbor-joining (NJ) method was employed and a phylogenetic tree with four (4) distinct clades was generated. Our isolates were found to be grouped in the same clade as the reference isolates of *S. vitis* (Napa764, L189, CBS 123004, P210) (Appendix A) with a strong bootstrap support of 100% (Appendix A). Furthermore, NJ analysis based on the combined sequences of *rDNA-ITS* and *LSU* showed that our *N. italica* isolate (SAROG1.3AVR7), along with the reference isolates (MFLU 14-0809 and 108) (Appendix A), formed a well-supported clade with a bootstrap value of 100% (Appendix A). Similar results were observed for the *K. variispora* isolates, as NJ analysis of the assembled sequences of *rDNA-ITS* and *LSU* demonstrated a well-supported clustering (93.4%) with the reference strains KV-9, CBS 121517, and KV-13 (Appendix A). Finally, we utilized combined sequences of *rDNA-ITS, LSU*, and *tub2* to construct a concatenated phylogenetic tree that included our isolate of *D. variabile* (SAROG1.2AKO1) along with other species of the genus *Didymospaeria*. The NJ analysis revealed a phylogenetic tree with two (2) well-supported clades, where our isolate (SAROG1.2AKO1) clustered together with the reference isolate *D. variabile* (CBS 120014) (Appendix A) with a bootstrap support of 100% (Appendix A).

Through blast and phylogenetic analysis, we successfully identified *N. italica* (SAROG1.3AVR7), *S. vitis* (SAROG1.3AVR10 and SAROG1.3AVR13), *D. variabile* (SAROG1.2AKO1), and *K. variispora* (HOURD2.1AVR1 and PEROG2.1YP2) isolates, and their colony color, growth rate, and microscopic features were indicative of their respective holotypes among the isolates.

### 3.6. Pathogenicity of Fungal Isolates

After a 6-month incubation period, longitudinal and transverse sections revealed severe wood tissue discoloration in inoculated canes treated with *S. vitis* (isolate SARO1.3AVR10), *D. variabile* (isolate SARO1.2AKO1), *K. variispora* (isolate HOURD2.1AVR1) and *N. italica* (isolate SARO1.3AVR7) compared to controls (Figure 10, Appendix A). Wood discoloration length caused by *N. italica* was significantly higher (*p* < 0.05) than the other fungal species tested. All these fungal species were successfully re-isolated from the artificially inoculated canes, whereas no fungus was obtained from control canes, thus confirming pathogenicity. No external leaf symptoms were observed in control and fungal-treated canes within the 6-month period of bioassays.

## 4. Discussion

In the last two decades, an increased incidence of grapevine trunk diseases (GTDs) has been reported in numerous studies, and their detrimental effect on the grapevine industry has raised awareness globally [4,7,10]. In the present study, a three-year survey (from 2018 to 2020) was carried out to investigate the incidence and identify fungi associated with GTDs in Greece. In total, 310 vineyards in different geographical regions with variable pedoclimatic conditions in northern, central, and southern Greece were surveyed, and 533 fungal strains were isolated from diseased vines and identified.

Measurements of GTDs’ incidence in geographically distinct regions in Greece revealed a higher frequency of symptomatic plants in vineyards in the warm and dry region of Heraklion (southern Greece) compared to those in cold and wet regions in northern Greece. Vineyards in central Greece (Nemea), characterized by moderate rainfall levels during the period of increased grapevine water requirements (in spring and summer months), indicated intermediate disease incidence. Differences in GTD symptom expression between vineyard regions can be attributed to variable climatic conditions [30]. Although correlation between climatic or environmental conditions and GTD symptom severity is not always clear, the combined effect of heat and water stresses on grapevine has been speculated to favor the initiation and progress of GTDs [10,11].

In addition, differences in the disease incidence between regions may be correlated with differences in the level of susceptibility to GTDs of the vine cultivars grown in each region. Knowledge of susceptibility for widely used plant genotypes is essential for estimating disease risk and planning effective management strategies [13]. The present study’s data suggest that grapevine cultivars grown in Greece are differentially affected by GTDs, thus demonstrating variable resistance. Considerably increased disease incidence (in terms of diseased vines and symptomatic/non-dead vines) was recorded on ‘Agiorgitiko’ grown primarily in central, and ‘Kotsifali’ and ‘Soultanina’ grown primarily in southern Greece. Apart from Kotsifali, these cultivars have been mentioned among the most susceptible to various GTD pathogens in several studies as defined via artificial inoculation experiments and field observations [31]. On the other hand, recent unpublished data of our group suggest that cultivars, such as Xinomavro or Limnio, which are among the most commonly used cultivars in northern Greece, are highly resistant to GTDs (Testempasis, unpublished data) and this could explain the lower disease incidence measured in northern Greece.

The occurrence of GTDs exhibits a positive correlation with the age of vines, displaying an exponential-related pattern. Our research findings indicate a heightened prevalence of GTDs in older vines (≥20 years old), followed by intermediate and lower incidence rates in middle-aged (10–19 years old) and younger (0–9 years old) vines, respectively. These outcomes align with earlier studies that observed increased frequencies of diseased vines in middle-aged and older categories compared to younger ones [32,33]. The variations in disease incidence across age groups were, to some extent, linked to the frequencies of GTD-inducing fungal genera isolated from each vine age category. Notably, the majority of GTD-inciting fungi were predominantly harbored in middle-aged and elder vines. Previous research has similarly documented a correlation between the vine’s age and the presence of specific GTDs, including White rot, Botryosphaeria and Eutypa dieback [7,34,35].

Climatic conditions, including heat and water stress, influence the prevalence and distribution of GTDs’ fungi [11]. Indeed, in the case of geographically distinct viticultural regions studied here, the population structure of GTD-related fungi differed significantly. Indicatively, the most predominant genera/species in regions of southern Greece (e.g., *P. chlamydospora* and *Diaporthe* spp.) were completely absent or found at very low frequencies in northern and/or central Greece and vice versa. These findings are strongly supported by a recent grapevine wood microbiome study conducted in Greece [14], pointing out that certain GTD-associated pathogens like *P. chlamydospora* and *Diaporthe* spp. were consistently present in cultivar ‘Vidiano’ collected from vineyards in southern (Heraklion, Crete) but not in ‘Xinomavro’ collected from vineyards in northern Greece (Amyntaio). On the contrary, *Diplodia* spp. (mostly *D. seriata*) were the most frequently isolated species in northern and central Greece but considerably less frequently in southern Greece. Furthermore, *Lasiodiplodia* spp. (mostly *L. viticola* and *L. theobromae*) were found in northern Greece, exclusively. Likewise, Hernandez and Alchala [32] reported higher abundancies of Botryosphaeria dieback pathogens in the northern regions of Oregon compared to the southern ones. Additionally, the elevated incidence of Botryosphaeria dieback complex pathogens in the northern regions of Greece might be associated with the prevalence of these pathogens in temperate regions or those with cold winters [36].

Out of 35 fungal genera identified in total, 25 were found in the warm and dry region in southern Greece, whereas 15 and 19 out of 35 genera were found in the comparatively cooler regions in northern and central Greece, respectively. Amongst them, 18 fungal genera are well-known (e.g., *Botryosphaeria dothidea*, *Diaporthe* spp., *Diplodia seriata*, *Eutypa* sp., *Fomitiporia mediterranea*, *Phaeoacremonium* spp., *Phaeomoniella clamydospora*) or lesser-known (i.e., *Fusarium* sp., *Seimatosporium* sp., *Didymosphaeria* sp., *Kalmusia* sp.) GTDs-inducing agents [7,16,19,20,21]. The remainigng 17 determined genera include wood canker agents or saprophytes (e.g., *Alternaria* sp., *Aspergillus* sp., *Penicillium* sp., *Rhizopus* sp.), endophytic symbionts with potential biocontrol activity (e.g., *Clonostachys* sp., *Epicoccum* sp., *Trichoderma* sp.), or genera that have not been reported previously as wood tissue colonizers (e.g., *Lopadostroma* sp., *Monocillium* sp., *Neosetophoma* sp.) [14,37,38,39,40,41]. The increased number of fungal genera originated from GTD-affected vines in southern Greece could be partially attributed to the thermophilic nature (within a certain limit) of most GTD-associated fungi [11]. In addition, the higher number of samples collected and analyzed from southern Greece may also play a role in the increased fungal diversity identified there.

The sequencing of *rDNA-ITS*, *LSU*, *tef1-α*, *tub2*, and *act* genes of selected isolates, coupled with pathogenicity tests, revealed that *N. italica*, *S. vitis*, *D. variabile* and *K. variispora* caused wood infection on grapevine canes, with the former species being the most virulent. Although these fungi were identified (at the genus level) in a grapevine wood microbiome study conducted recently, their pathogenic potential was not examined [14]. Therefore, this is the first record of *K. variispora*, *S. vitis* and *D. variabile* associated with GTDs in Greece. *Seimatosporium vitis* has never been isolated in Greece previously, whereas *K. variispora* and *D. variabile* have been reported as causal agents of fruit rot on apples [42] and leaf spot on *Phoenix theophrasti* [43], respectively. Τhis is also the first report of *N. italica* associated with GTDs worldwide. *Neosetophoma italica* was first found as saprobic on dead leaves of *Iris germanica* L. in Italy and typified in 2015 [44]. Its close relative, *N. samararum* has been reported as a pathogen causing leaf spots of various hosts [45], but to date, *N. italica* has never been shown to infect plant hosts. The increased virulence of *N. italica* revealed here may suggest the predominant role of this species in GTD-associated microbiome, and should be taken into consideration in future surveys in Greece and elsewhere.

Given all the above, the data from the present study suggest an overall increased incidence of GTDs in Greece. Differences in disease incidence and population structure of the GTD-associated microbiome were observed among the discrete viticultural zones. The increased incidence of GTDs and the high variability in fungal genera implicated with the disease complex recorded in the driest and warmest region of Heraklion (southern Greece) may portend even more ominous scenarios for viticulture sustainability globally against climate change. Apart from the different climatological conditions, these variances may be reasonably attributed to the differential susceptibility level of grapevine genotypes cultivated in geographically distinct viticultural regions. Here, we record, for the first time, *K. variispora*, *S. vitis* and *D. variabile* as GTD-associated pathogens in Greece, and *N. italica* as GTDs-inducing pathogen worldwide. This highly virulent species may have a predominant role in GTDs’ complex, and its biological cycle and epidemiology should be further investigated. Regional-scale mapping conducted here will provide valuable support in planning and implementing effective management practices targeting specific diseases.

## Figures and Tables

**Figure 1 jof-10-00002-f001:**
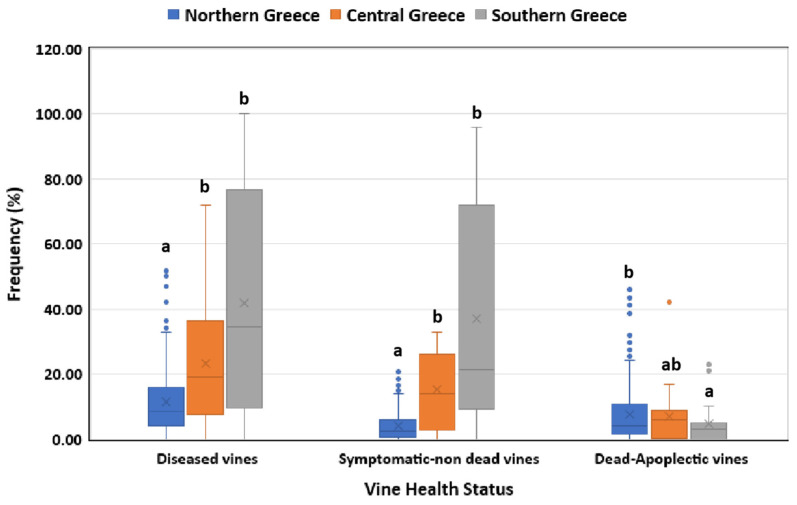
Grapevine trunk disease incidence (symptomatic/non-dead and dead/apoplectic vines) in vineyards located in northern, central and southern Greece. Within the box, the median and the mean are represented by the solid line and the ‘X’, respectively. Top and bottom lines of the box correspond to the 25th and 75th percentiles of the data, respectively. Error bars represent the 10th and 90th percentiles. Within each group of plants, columns with the same letter do not differ significantly according to Kruskal–Wallis and Dunn’s test for multiple comparisons (*p* ≤ 0.05).

**Figure 2 jof-10-00002-f002:**
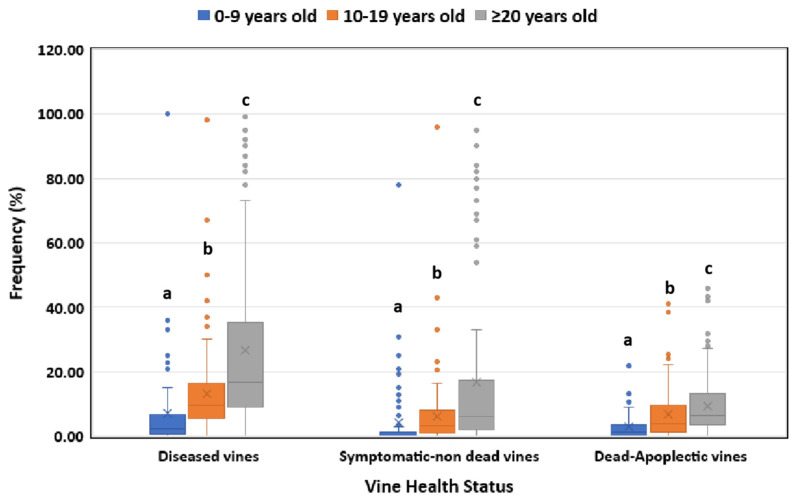
Grapevine trunk disease incidence (symptomatic/non-dead and dead/apoplectic vines) for young (0–9 years old), middle-aged (10–19 years old) and elder (≥20 years old) vineyards. Within the box, the median and the mean are represented by the solid line and the ‘X’, respectively. Top and bottom lines of the box correspond to the 25th and 75th percentiles of the data, respectively. Error bars represent the 10th and 90th percentiles. Within each group of plants, columns with the same letter do not differ significantly according to Kruskal–Wallis and Dunn’s test for multiple comparisons (*p* ≤ 0.05).

**Figure 3 jof-10-00002-f003:**
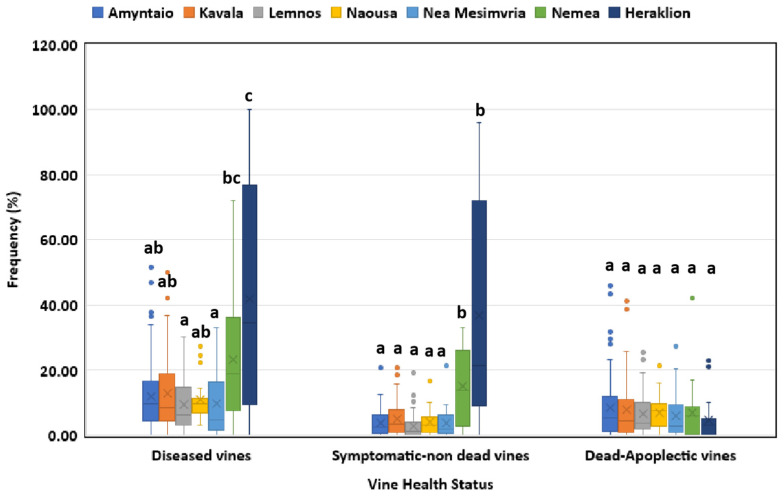
Grapevine trunk disease incidence (diseased, symptomatic/non-dead and dead/apoplectic vines) for vineyards located in different regions in Greece. Within the box, the median and the mean are represented by the solid line and the ‘X’, respectively. Top and bottom lines of the box correspond to the 25th and 75th percentiles of the data, respectively. Error bars represent the 10th and 90th percentiles. Within each group of plants, columns with the same letter do not differ significantly according to Kruskal–Wallis and Dunn’s test for multiple comparisons (*p* ≤ 0.05).

**Figure 4 jof-10-00002-f004:**
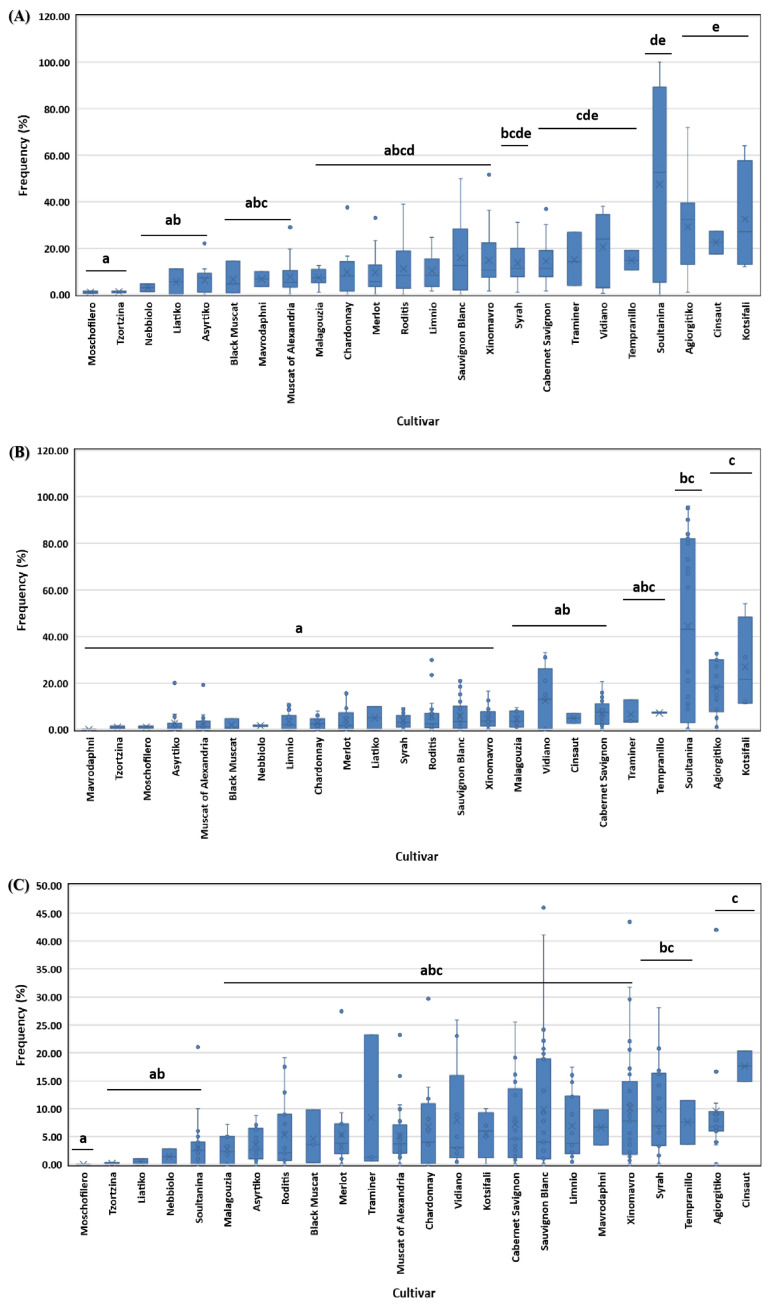
Frequency (%) of diseased (**A**), symptomatic/non-dead (**B**) and dead/apoplectic (**C**) vines on 24 grapevine cultivars grown in Greece. Within the box, the median and the mean are represented by the solid line and the ‘X’, respectively. Top and bottom lines of the box correspond to the 25th and 75th percentiles of the data, respectively. Error bars represent the 10th and 90th percentiles. Columns with the same letter do not differ significantly according to Kruskal–Wallis and Dunn’s test for multiple comparisons (*p* ≤ 0.05).

**Figure 5 jof-10-00002-f005:**
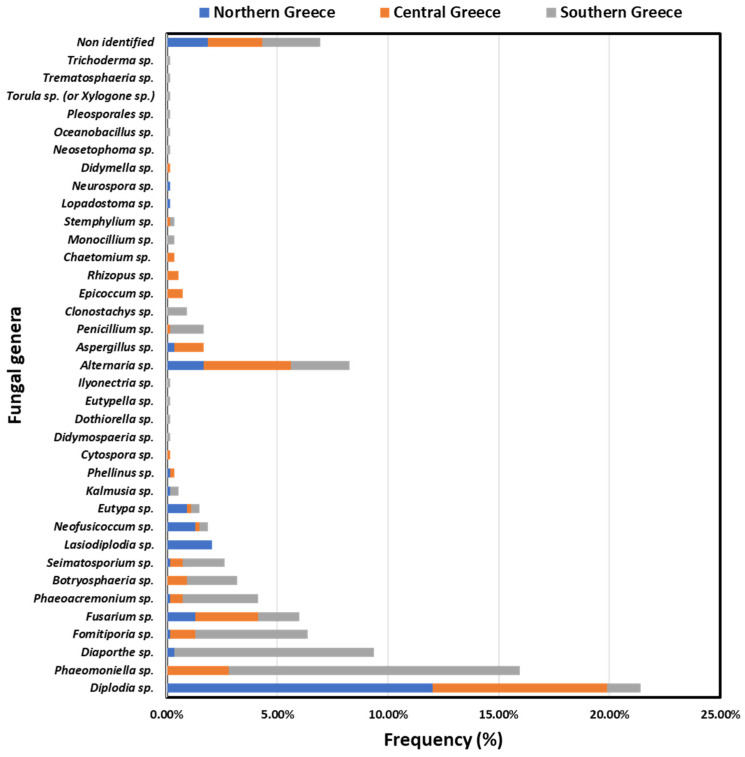
Frequencies of isolates belonging to discrete fungal genera (and order) in northern (Amyntaio and Kavala), central (Nemea) and southern (Heraklion) Greece.

**Figure 6 jof-10-00002-f006:**
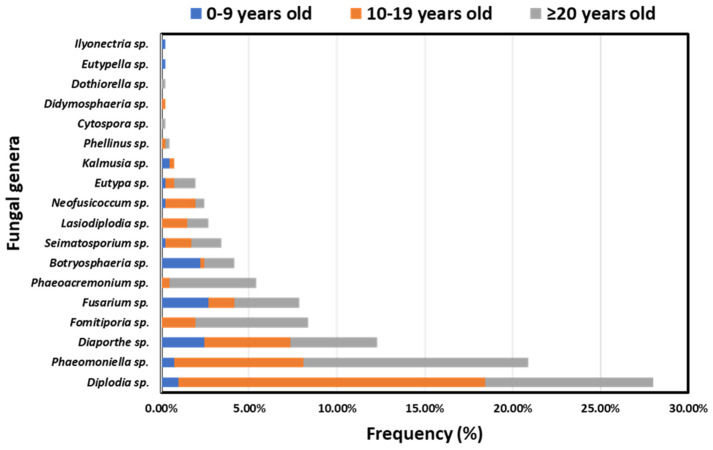
Frequencies of isolates belonging to 18 discrete fungal genera associated with grapevine trunk diseases, originating from young (0–9 years old), middle-aged (10–19 years old) and elder (≥20 years old) vineyards in Greece.

**Figure 7 jof-10-00002-f007:**
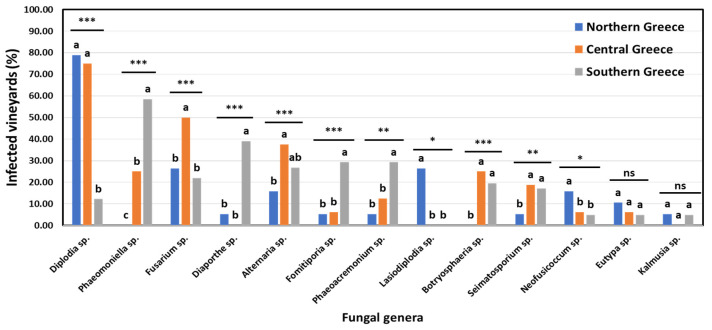
Percentage of vineyards infested by each fungal genera in northern (Amyntaio and Kavala), central (Nemea) and southern (Heraklion) Greece. Within each genus, columns with the same letter do not differ significantly according to Chi-square test and a pair-wise comparison, whereas asterisks (*, ** and ***) indicate significance at *p* ≤ 0.05, 0.01 and 0.001, respectively. ‘ns’ indicates no significance.

**Figure 8 jof-10-00002-f008:**
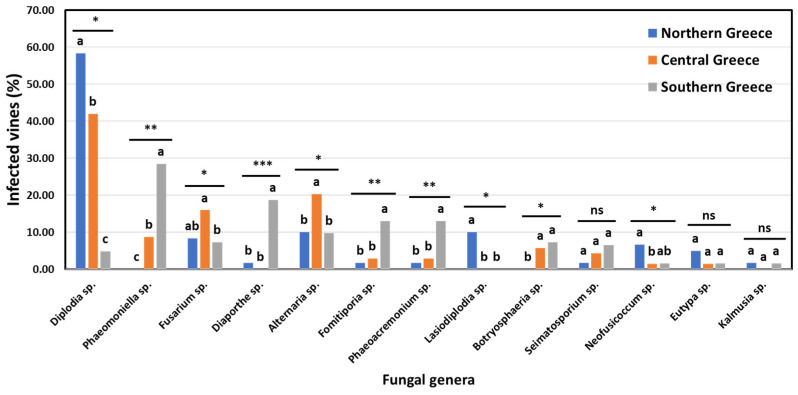
Percentage of vines infested by each fungal genera in northern (Amyntaio and Kavala), central (Nemea) and southern (Heraklion) Greece. Within each genus, columns with the same letter do not differ significantly according to Chi-square test and a pair-wise comparison, whereas asterisks (*, ** and ***) indicate significance at *p* ≤ 0.05, 0.01 and 0.001, respectively. ‘ns’ indicates no significance.

**Figure 9 jof-10-00002-f009:**
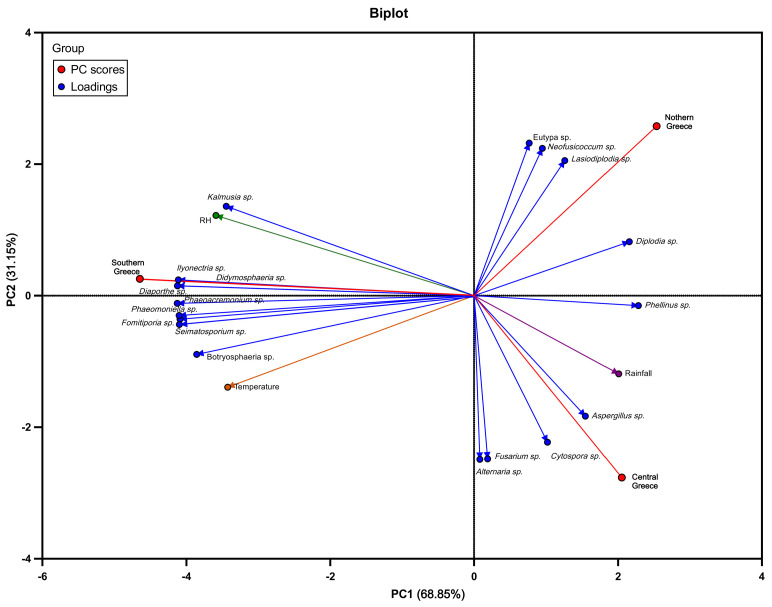
Principal component analysis (PCA) biplot illustrating the correlation between meteorological data (■ air temperature, ■ RH, and ■ rainfall) from past decades (2010–2020) and the identified fungal genera isolated from symptomatic grapevines in the viticultural zones of southern, central, and northern Greece.

**Figure 10 jof-10-00002-f010:**
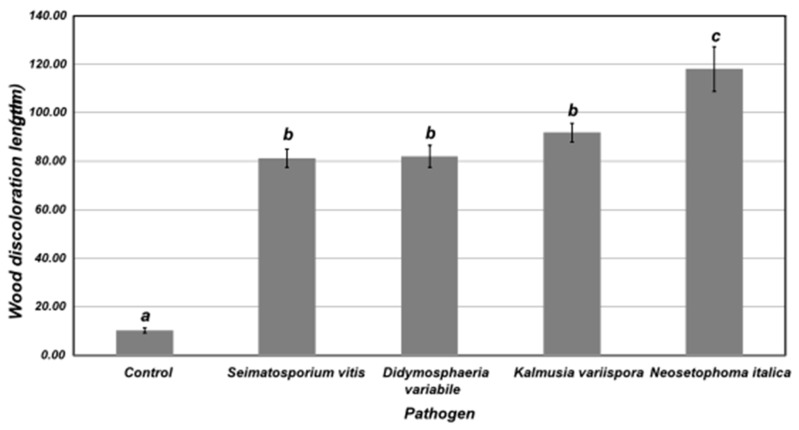
Length of vascular tissue discoloration in 2-year-old canes of cv. Soultanina 6 months after artificial inoculation with *Seimatosporium vitis* (isolate SARO1.3AVR10), *Didymosphaeria variabile* (isolate SARO1.2AKO1), *Kalmusia variispora* (isolate HOURD2.1AVR1) and *Neosetophoma italica* (isolate SARO1.3AVR7). Control canes (C-) were mock-inoculated with sterilized PDA discs. Columns followed by the same letter do not differ significantly according to Tukey significant difference test (*p* ≤ 0.05). Each column represents the mean of 20 canes and vertical bars indicate standard errors.

**Table 1 jof-10-00002-t001:** Numerical distribution of vineyards located in seven (7) important grapevine-growing regions in three (3) geographically distinct viticultural zones (northern, central and southern Greece) falling into three (3) age-scales, surveyed for the assessment of GTD disease parameters.

Viticultural Zone	Region	Number of Vineyards	Vineyard Age (Years)		
			<10	10–19	≥20
Northern Greece	Amyntaio	78	10	39	29
	Kavala	60	8	27	25
	Lemnos	51	21	14	16
	Naousa	20	0	17	3
	Nea Mesimvria	25	9	7	9
Central Greece	Nemea	28	3	8	17
Southern Greece	Heraklion	48	19	4	25
Total		310	70	116	124

## Data Availability

Data are contained within this article.

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
