# Peer review of "Grapevine Trunk Diseases in Greece: Disease Incidence and Fungi Involved in Discrete Geographical Zones and Varieties"

_jof, 2023, doi:10.3390/jof10010002_

Round 1
Reviewer 1 Report
Comments and Suggestions for Authors
The submitted article is within the scope of the journal, is properly substantiated and structured and the results are scientifically relevant, however some aspects need to be clarified and discussed.
In the section 2.1 Field surveys and disease incidence assessment: at what time of year or stage of phenological development of the vineyard were the samples collected?
In the section 2.3 Morphological and cultural characterization of isolates, line 152: reproductive structures instead of carpophores, which is a term commonly used for macrofungi
In the section 2.4 DNA extraction, PCR amplification and sequencing, line 161 Fungal DNA was extracted…: although Cary et al. [23] is refered, present a brief description of the methodology of DNA extraction like using kit X from brand Y, or the name of the method.
In the section 2.5 Identification and characterization of isolates, line 199: evolutionary analyses were conducted by employing the Neighbor Joining method (NJ) and Tamura-Nei model [29]. - This method of phylogenetic analysis is not statistically robust. Why your choice? Why not use maximum likelihood?
In the section 2.7 Pathogenicity tests: Only one isolate of each species was used. Why? Discuss limitations.
In the results the order of the figures and paragraphs in which they are referred must be followed so that we do not have to move back and forth in the article (even if figures appear in a row).
The concepts of disease assessment in the field have to be clarified: diseased vines (including dead and non-dead vines), symptomatic but non-dead vines, dead/apoplectic vines - is diseased vines the sum of symptomatic but non-dead vines and dead/apoplectic vines? It is not reflected in the graphics. Why duplicate information?
Figure 5 - Pleosporales is not a genus is an order of fungi
Figure 9 – meteorological vectors should be in a different color too
In the list of bibliographic references, the names of species and genera must be in italics.
Correct the titles of references 8 and 9.
Author Response
Dear Reviewer 1,
in the following paragraphs we address our line by line responses to your comments:
- In the section 2.1 Field surveys and disease incidence assessment: at what time of year or stage of phenological development of the vineyard were the samples collected?
RESPONSE
In the section 2.2 Sampling and fungal isolation: we added ‘During the surveys (in July and August of the years 2018-2020), wood samples…’
- In the section 2.3 Morphological and cultural characterization of isolates, line 152: reproductive structures instead of carpophores, which is a term commonly used for macrofungi.
RESPONSE
We replaced ‘carpophores’ with ‘reproductive structures’ as the reviewer 1 suggested.
- In the section 2.4 DNA extraction, PCR amplification and sequencing, line 161 Fungal DNA was extracted…: although Cary et al. [23] is refered, present a brief description of the methodology of DNA extraction like using kit X from brand Y, or the name of the method.
RESPONSE
A brief description of the method was added, according to reviewer’s instructions.
- In the section 2.5 Identification and characterization of isolates, line 199: evolutionary analyses were conducted by employing the Neighbor Joining method (NJ) and Tamura-Nei model [29]. - This method of phylogenetic analysis is not statistically robust. Why your choice? Why not use maximum likelihood?
RESPONSE
Our choice of the Neighbor Joining (NJ) method with the Tamura-Nei model for evolutionary analyses was influenced by several factors specific to our study. Firstly, the inclusion of multi-locus sequences posed significant computational challenges, and the NJ method's computational efficiency allowed us to manage these complexities. Secondly, preliminary analyses demonstrated that the NJ method produced tree topologies in line with our expectations and consistent with prior research in the field, providing satisfactory congruence. Furthermore, our primary aim for the phylogenetic analysis was to complement and support our identification based on blast analysis. We acknowledge the increased statistical robustness associated with maximum likelihood (ML) methods and are open to conducting ML analyses if the reviewer deems it crucial for further validation.
In the section 2.7 Pathogenicity tests: Only one isolate of each species was used. Why? Discuss limitations.
RESPONSE
Indeed, we used one isolate per species, since our purpose was to test the pathogenic potential of these isolates on grapevine instead of evaluating the virulence of different isolates of the same species. Then, we noticed that the isolate SARO1.3AVR7 (Neosetophoma italica) caused increased wood symptoms compared to the other 3 isolates tested.
- In the results the order of the figures and paragraphs in which they are referred must be followed so that we do not have to move back and forth in the article (even if figures appear in a row).
RESPONSE
We transferred the paragraphs describing the results so that they are followed by the respective figures they are referred to (Figures 3, 4, 6 and 8).
- The concepts of disease assessment in the field have to be clarified: diseased vines (including dead and non-dead vines), symptomatic but non-dead vines, dead/apoplectic vines - is diseased vines the sum of symptomatic but non-dead vines and dead/apoplectic vines? It is not reflected in the graphics. Why duplicate information?
RESPONSE
It may seem a bit confusing. Indeed, diseased vines is the sum of symptomatic but non-dead vines plus dead/apoplectic vines. Please, notice that within each box, the mean of each disease parameter is indicated with ‘X’. For instance, in Figure 1, in the case of Southern Greece (grey box), the mean value for Diseased vines is 41.69%, equal with the sum of the mean value of Symptomatic-non dead vines (36.90%) plus the mean value of Dead-Apoplectic vines (4.79%). So, we think that the concepts of disease assessment in the field is clearly reflected in the graphics, based on the individual diseased parameters presented here. In respect of the duplicate information that Reviewer 1 claim, we believe that individual disease parameters may provide a better discrimination among different cases, leading in more precise conclusions. For example, in Figure 1 again, the frequency of diseased vines in Southern Greece is significantly higher than in Northern Greece and the same seems for Symptomatic-non dead vines. However, vineyards in Southern Greece, exhibited significantly lower frequency of dead-apoplectic vines compared to the ones in Northern Greece. Please, see also the results in figure 3 …
- Figure 5 - Pleosporales is not a genus is an order of fungi
RESPONSE
REVIEWER 1 is correct!!! Therefore, we added ‘(and order) in Figure 5 caption.
- Figure 9 – meteorological vectors should be in a different color too
RESPONSE
The reviewer's suggestions regarding color modifications were accommodated, and the figure has been updated accordingly.
In the list of bibliographic references, the names of species and genera must be in italics.
RESPONSE
Species and genera were changed in Italics
- Correct the titles of references 8 and 9.
RESPONSE
The titles of the references 8 and 9 were corrected.
Reviewer 2 Report
Comments and Suggestions for Authors
Congratulations to all the authors for their excellent work on Grapevine trunk diseases in Greece: Disease incidence and fungi involved in discrete geographical zones and varieties.
The authors conducted a three-year survey to estimate the incidence of grapevine trunk diseases (GTDs) in Greece and identify fungi associated with the disease complex. In total, 310 vineyards in different geographical regions in northern, central, and southern Greece were surveyed, and 533 fungal strains were isolated from diseased vines. Morphological, physiological and molecular (5.8S rRNA gene-ITS sequencing) analyses revealed that isolates belonged to 35 distinct fungal genera, including well-known (e.g. Botryosphaeria sp, Diaporthe spp., Eutypa sp., Diplodia sp., Fomitiporia sp., Phaeoacremonium spp., Phaeomoniella sp.) and lesser-known (e.g. Neosetophoma sp., Seimatosporium sp., Didymosphaeria sp., Kalmusia sp.) grapevine wood inhabitants. GTDs-inducing population structure differed significantly among the discrete geographical zones. Phaeomoniella chlamydospora (26.62%, n=70), Diaporthe spp. (18.25%, n=48) and F. mediterranea (10.27%, n=27) were the most prevalent in Heraklion, whereas D. seriata, Alternaria spp., P. chlamydospora and Fusarium spp. were predominant in Nemea (central Greece). In Amyntaio and Kavala (northern Greece), D. seriata was the most frequently isolated species (>50% frequency). Multi-genes (rDNA-ITS, LSU, tef1-α, tub2, act) sequencing of selected isolates, followed by pathogenicity tests revealed that Neosetophoma italica, Seimatosporium vitis, Didymosphaeria variabile and Kalmusia variispora caused wood infection with the former being the most virulent.
In this study they also report for the first time K. variispora, S. vitis and D. variabile as GTD-associated pathogens in Greece, and N. italica as GTDs-inducing pathogen worldwide. This highly virulent species may have a predominant role in GTDs complex, and its biological cycle and epidemiology should be further investigated. Regional-scale mapping conducted in this work will provide valuable support in planning and implementing effective management practices targeting specific GTDs in Greece and also worldwide.
Author Response
Dear Reviewer 2,
in the following paragraphs we address our line by line responses to your comments:
- Congratulations to all the authors for their excellent work on Grapevine trunk diseases in Greece: Disease incidence and fungi involved in discrete geographical zones and varieties.
RESPONSE
We all thank the reviewer 2 for the very kind compliments.
Reviewer 3 Report
Comments and Suggestions for Authors
Dear authors and editor,
the manuscript “Grapevine trunk diseases in Greece: Disease incidence and fungi involved in discrete geographical zones and varieties” reports a three-year analysis of the incidence of grapevine trunk diseases in Greece and the fungi associated with the disease.
The manuscript is well written and enjoyable to read. The study is complex and comprehensive and the results shown in the graphs helps to characterise the incidence of GTDs in Greece and the drivers of the disease. The study may represent a study for implementing managing strategies in the country.
Before considering the publication of the work, there are some points to clarify:
- The number of vineyards surveyed in each viticultural area are widely different. Could you clarify if the number of vineyards is proportional to the cultivated area?
- What’s the difference between non-dead diseased vines and symptomatic non-dead vines?
- What do you mean by disease indices?
- Tempranillo must be correctly spelled (line 299)
Author Response
Dear Reviewer 3,
In the following paragraphs we address our point-by point response to your comments:
Reviewer 3
- The number of vineyards surveyed in each viticultural area are widely different. Could you clarify if the number of vineyards is proportional to the cultivated area?
RESPONSE
Acknowledging the reviewer's observation regarding the uneven distribution of surveyed vineyards among viticultural areas, we would like to inform you that the discrepancy arises from the diverse viticultural landscape in Northern Greece. This region encompasses multiple viticultural zones with significant divergence, leading to a higher sampling rate to obtain more reliable results in terms of disease parameters and fungal diversity involved in GTDs complex. Conversely, in central and southern Greece, where viticultural zones are more confined to specific regions without substantial divergence, fewer vineyards were sampled.
- What’s the difference between non-dead diseased vines and symptomatic non-dead vines?
RESPONSE
There is no difference, it is the same.
- What do you mean by disease indices?
RESPONSE
By ‘disease indices’ we mean ‘disease parameters’.…
- Tempranillo must be correctly spelled (line 299)
RESPONSE
‘Tempranilo’ was corrected to ‘Tempranillo’, as the Reviewer 3 suggested.